# Cancer Takes a Toll on Skeletal Muscle by Releasing Heat Shock Proteins—An Emerging Mechanism of Cancer-Induced Cachexia

**DOI:** 10.3390/cancers11091272

**Published:** 2019-08-30

**Authors:** Thomas K Sin, Guohua Zhang, Zicheng Zhang, Song Gao, Min Li, Yi-Ping Li

**Affiliations:** 1Department of Integrative Biology and Pharmacology, The University of Texas Health Science Center at Houston, 6431 Fannin Street, Houston, TX 77030, USA; 2The Vivian L. Smith Department of Neurosurgery, The University of Texas Health Science Center at Houston, Houston, TX 77030, USA; 3Department of Medicine, The University of Oklahoma Health Sciences Center, Oklahoma City, OK 73104, USA; 4Department of Surgery, The University of Oklahoma Health Sciences Center, Oklahoma City, OK 73104, USA

**Keywords:** cancer, muscle wasting, TLR4, Hsp70, Hsp90

## Abstract

Cancer-associated cachexia (cancer cachexia) is a major contributor to the modality and mortality of a wide variety of solid tumors. It is estimated that cachexia inflicts approximately ~60% of all cancer patients and is the immediate cause of ~30% of all cancer-related death. However, there is no established treatment of this disorder due to the poor understanding of its underlying etiology. The key manifestations of cancer cachexia are systemic inflammation and progressive loss of skeletal muscle mass and function (muscle wasting). A number of inflammatory cytokines and members of the TGFβ superfamily that promote muscle protein degradation have been implicated as mediators of muscle wasting. However, clinical trials targeting some of the identified mediators have not yielded satisfactory results. Thus, the root cause of the muscle wasting associated with cancer cachexia remains to be identified. This review focuses on recent progress of laboratory studies in the understanding of the molecular mechanisms of cancer cachexia that centers on the role of systemic activation of Toll-like receptor 4 (TLR4) by cancer-released Hsp70 and Hsp90 in the development and progression of muscle wasting, and the downstream signaling pathways that activate muscle protein degradation through the ubiquitin–proteasome and the autophagy–lysosome pathways in response to TLR4 activation. Verification of these findings in humans could lead to etiology-based therapies of cancer cachexia by targeting multiple steps in this signaling cascade.

## 1. Introduction

Cancer has been increasingly recognized as a systemic disorder that impairs multiple organs in which cancer may or may not reside. Cachexia is a frequent complication of cancer seen in ~60% of all cancer patients [1]. Cachexia is defined as a multifactorial syndrome characterized by an ongoing loss of skeletal muscle mass (with or without loss of fat mass) that cannot be fully reversed by conventional nutritional support and leads to progressive functional impairment [2]. The primary clinical manifestations of cachexia are weight loss, inflammation, insulin resistance, and increased muscle protein breakdown. The progressive loss of muscle mass (muscle wasting) causes systemic disorders in body metabolism, decreases the efficacy while increasing the toxicity of chemotherapy, and accelerates the physical deterioration and the demise of cancer patients [3]. Consequently, cachexia is estimated as the immediate cause of one third of all cancer-related death [1,2,4]. However, there is no FDA-approved treatment for cancer cachexia due to the lack of a clearly defined etiology of this lethal disorder. There is a clearly unmet medical need to improve our understanding of the cause of cancer cachexia, and devise therapeutic strategies accordingly.

Although muscle wasting in cancer hosts results from multiple alterations in the homeostasis of skeletal muscle including decreased protein synthesis and myogenic regeneration (anabolism) as well as increased protein breakdown (catabolism), the latter is considered to be the primary contributor to muscle wasting. In particular, accelerated muscle protein degradation due to hyperactivation of the cellular protein degradation machineries including the ubiquitin-proteasome pathway (UPP) and the autophagy-lysosomal pathway (ALP) is considered central to muscle wasting. The UPP degrades myofibrillar proteins and regulatory proteins through substrate-specific E3s, while autophagy involves autophagosomal engulfment of cytoplasmic constituents including ubiquitinated protein aggregates and organelles for degradation by lysosomes [5]. However, despite the identification of more than a dozen of humoral factors including specific inflammatory cytokines and members of the TGFβ superfamily that may contribute to the accelerated muscle protein breakdown in cancer cachexia, the key mechanisms that initiate and sustain cancer cachexia remained elusive. This situation prompted the National Cancer Institute to issue the Provocative Question in 2012 and 2013: What mechanisms initiate or sustain cancer cachexia, and can we target them to extend lifespan and quality of life for cancer patients? (RFA-CA-12-021 and RFA-CA-13-018). Intense research efforts have led to the identification of novel pathogenic mechanisms of cancer-induced muscle wasting. This review summarizes the recent findings from laboratory models of cancer cachexia that the release of extracellular vesicle-associated Hsp70 and Hsp90 by cancer activates Toll-like receptor 4 (TLR4) that provokes muscle catabolism by (1) direct activation of UPP and ALP in skeletal muscle cells through specific signaling pathways and (2) induction of systemic inflammation leading to the release of cytokines that also stimulate muscle protein degradation pathways. These data shifted the existing paradigm for the mechanisms that initiate and sustain cancer cachexia and, if confirmed in humans, would lead to new therapeutic interventions for muscle wasting associated with cancer cachexia.

## 2. Inflammatory Signaling Cascades Mediate Skeletal Muscle Protein Degradation in the Cancer Milieu

Data from animal models revealed that skeletal muscle atrophy induced by diverse stimuli share a common mechanism, for example, accelerated muscle protein degradation results primarily from a coordinated activation of the UPP and ALP. However, the underlying signaling mechanisms that initiate and sustain the catabolic process in different pathological conditions vary. In fasting, denervation or disuse-induced muscle atrophy, muscle catabolism is mediated by upregulation of rate-limiting E3 ubiquitin ligases atrogin1/MAFbx and MuRF1 [6,7]. MuRF1 directly targets myofibrillar proteins including myosin and α-actin for ubiquitylation and degradation by the 20S proteasome [8,9,10]. Instead of directly targeting myofibrillar proteins, atrogin1/MAFbx targets regulatory proteins such as eukaryotic initiation factor 3 subunit 5 (eIF3-f) [11,12] and MyoD [13]. Both atrogin1/MAFbx and MuRF1 are upregulated by FoxO1/3 transcription factors, which is inversely regulated by PI3K-Akt signaling in response to IGF-I activity [14,15]. The PI3K-Akt-FoxO1/3 signaling pathway also inversely regulates autophagy activity in response to fasting, denervation or disuse by regulating autophagy-related genes including *LC3* and *Bnip3* [16]. In fact, the PI3K-Akt-FoxO1/3 signaling pathway coordinately activates protein degradation by both the UPP and ALP in denervation or fasting [17,18]. Cancer-induced muscle wasting is also mediated by the common protein degradation machineries including the UPP and ALP, which are similarly activated in the cachectic muscle of tumor-bearing mice [19,20,21,22,23] and cancer patients [24,25,26]. However, activation of the UPP and ALP in the muscle of cancer hosts does not appear to require the PI3K-Akt-FoxO1/3 signaling pathway. In fact, Akt is activated in the cachectic muscle of tumor-bearing mice [27,28] and cancer patients [25,26], which inhibits FoxO1/3, resulting in decreased activity of UPP and ALP [14,15]. Thus, the signaling mechanisms mediating the activation of UPP and ALP in skeletal muscle by cancer are distinct from those by fasting, denervation, or disuse.

Cancer cachexia is characterized by systemic inflammation, which is absent in muscle atrophy induced by fasting, denervation, or disuse. TNFα, also known as cachectin, is the first inflammatory cytokine linked to cancer cachexia due to its elevation in the circulation of cancer patients with cachexia, and its capacity to induce muscle wasting in laboratory animals [29]. Other cytokines including IL-6 [30,31], IL-1β [32], leukemia inhibitory factor (LIF) [33], and TNF-like weak inducer of apoptosis (TWEAK) [34] as well as members of the TGFβ superfamily including activin A [35], myostatin [36], TGFβ [37], and GDF11 [38] were subsequently shown to be involved in the promotion of muscle catabolism in animal models of cancer cachexia. Similar to cancer, many of these factors including TNFα, IL-6, IL-1, LIF, activin A, and myostatin activate Akt in skeletal muscle cells while stimulating muscle protein loss [32,39,40]. These catabolic factors activate muscle protein degradation through inflammatory signaling molecules including NF-κB [41,42], p38 mitogen-activated protein kinase (MAPK) [39,43], and STAT3 [31,44]. Although some types of cancer cells can release certain cytokines, cachectic cancer cells do not necessarily release catabolic cytokines such as TNFα, IL-6, and IL-1β, and the vast majority of circulating cytokines are generated by immune cells in response to cancer [45]. Cytokines act as a network to amplify inflammation. Thus, targeting individual cytokines may not be highly effective for intervening cancer cachexia. In fact, clinical intervention of cancer cachexia using anti-cytokine strategies did not yield satisfactory results [46,47]. Therefore, identifying and targeting the root cause of cancer-induced inflammation is necessary for the successful intervention of cancer-induced muscle wasting. Indeed, emerging evidence has revealed the potential root cause of cancer-induced inflammation and cachexia.

## 3. TLR4 Activation in Muscle Cells Causes Muscle Wasting

Toll-like receptors are pattern recognition receptors that are important mediators of innate immunity [48] and are involved in host responses to cancer [49]. The role of TLR4 in muscle wasting was initially revealed by Cannon and colleagues [50]. Utilizing C3H/HeJ mice that have nonfunctional TLR4 due to a double mutation, they observed that intact TLR4 is required for muscle wasting induced by engrafted SCCF-VII tumor cells. Based on the observation that tumor failed to induce IL-1β elevation in C3H/HeJ mice, the authors proposed that cancer induced muscle wasting through TLR4-mediated systemic inflammatory. McClung et al. [51] reported that administration of lipopolysaccharide (LPS), the classical ligand of TLR4, to mice upregulates autophagy-related genes (Atg6, Atg7, and Atg12) in skeletal muscle. Given that TLR4 is expressed by skeletal muscle cells, Doyle et al. [52] postulated that TLR4 activation in muscle cells directly stimulates muscle catabolism independent of its activation of systemic inflammation. They demonstrated for the first time that LPS-treated C2C12 myotubes undergo rapid loss of muscle proteins due to p38 MAPK-mediated activation of both UPP and ALP. Importantly, they showed that TLR4 knockout mice are resistant to LPS-induced muscle catabolism. This study revealed for the first time that TLR4 activation in skeletal muscle cells is sufficient to cause muscle catabolism, independent of the systemic release of cytokines.

Although certain types of cancer that cause gut barrier dysfunction can lead to endotoxemia [53], resulting in systemic activation of TLR4, endotoxemia is not a common feature of cancer. On the other hand, a number of endogenous TLR4 agonists have been identified and recognized as danger associated molecular patterns (DAMPs) including heat shock proteins, HMGB1, and angiotensin II [54] that recapitulate the effects of LPS. Therefore, Zhang and colleagues [22] tested the hypothesis that endogenous TLR4 agonists induce the muscle wasting associated with cancer cachexia. Utilizing TLR4-deficient myotubes and TLR4 knockout mice, they demonstrated that Lewis lung carcinoma (LLC) cells release unidentified factors that directly activate TLR4 and its downstream p38 MAPK–C/EBPβ and NF-κB signaling pathways in skeletal muscle cells to induce muscle catabolism mediated by the UPP and ALP, which is in congruent with the established effects of LPS [52]. In addition, LLC induces elevation of serum inflammatory cytokines (TNFα and IL-6) in a TLR4-dependent manner. This study revealed that LLC causes muscle wasting by releasing endogenous TLR4 agonists to activate the catabolic pathways in skeletal muscle cells directly, and activate the release of inflammatory cytokines that promote muscle catabolism indirectly. Not only do these data explain why anti-cytokine strategies do not effectively intervene in cancer cachexia, but also bring forward the translational implication of TLR4 inhibition in the clinical management of cancer cachexia. Recent corroborating studies have revealed that the two TLR4 adaptor proteins MyD88 and TRIF mediate LLC, C26, and pancreatic cancer-induced muscle wasting [55,56,57]. However, muscle-specific deletion of MyD88 does not protect against muscle protein loss induced by LPS [58], suggesting a difference between cancer and LPS-induced muscle wasting.

Interestingly, the role of TLR4 in cancer-induced muscle catabolism appears relatively specific. TLR4, but not TLR2, is required for LLC-induced muscle catabolism [22]. On the other hand, activation of TLR7 mediates skeletal muscle cell death induced by LLC and some human cancer cells [59]. Although TLR4 expression in skeletal muscle cells is not altered by LLC [22], a study of cancer patients found that TLR4 expression level in skeletal muscle is significantly correlated with low skeletal muscle index and weight loss [60]. In addition, increased TLR4 expression has been observed in cardiac myocytes of heart failure patients [61,62], who frequently develop skeletal muscle atrophy. Furthermore, TLR4 is activated in the skeletal muscle of patients with chronic kidney disease that frequently causes muscle wasting [63]. In contrast, disuse-induced muscle atrophy is independent of TLR4 [64]. These data suggest that TLR4 mediates muscle catabolism in a number of chronic diseases that feature inflammation, but not in muscle atrophy induced by non-inflammatory conditions.

Systemic activation of TLR4 by cancer can alter the homeostasis of other organs. Henriques et al. reported that LLC tumor-bearing TLR4 knockout mice are resistant to adipocyte atrophy and macrophage infiltration in adipose tissue, suggesting a role of TLR4 in mediating cancer-induced adipose tissue loss [65]. Given that TLR4 is expressed in a wide range of cells, it is conceivable that cancer causes systemic disorders by activating TLR4 in many organs.

## 4. Intracellular Signaling Pathways that Mediate TLR4-Induced Muscle Catabolism

TLR4 activation leads to the activation of downstream signaling molecule TRAF6, which in turn activates p38 MAPK and NF-κB [66] that are known to mediate muscle catabolism induced by cytokines or oxidative stress [43,67]. To identify TLR4-activated signaling molecules that mediate muscle catabolism, Doyle et al. showed that a p38α/β MAPK inhibitor (SB202190) abrogates LPS-induced muscle catabolism by inhibiting the activation of both the UPP and ALP in myotubes and in mice [52]. The p38 MAPK family is comprised of four isoforms, three of which (α, β, and γ) are expressed in skeletal muscle with distinct functions: p38α promotes myogenesis [68,69], whereas p38γ promotes satellite cell self-renewal [70] and other muscle activities [71,72]. To identify the p38 MAPK isoform that regulates muscle catabolism, Zhang et al. [28] found that p38β MAPK specifically upregulates atrogin1/MAFbx through C/EBPβ-mediated gene transcription. In addition, C/EBPβ knockout mice are resistant to LLC-induced muscle catabolism, suggesting that the p38β MAPK−C/EBPβ signaling pathway mediates the loss of myofibrillar proteins in response to a tumor burden. Further studies revealed that p38β MAPK activates the binding of C/EBPβ to the *atrogin1/MAFbx* gene promoter due to its phosphorylation of the Thr-188 residue of C/EBPβ [73].

The p38β MAPK–C/EBPβ signaling pathway also mediates the upregulation of the E3 ligase UBR2 in cachectic muscle of LLC tumor-bearing mice [74]. In addition, UBR2 is upregulated in C26 tumor-bearing mice and YAH-130 tumor-bearing rats [75,76], suggesting a role for this E3 in muscle catabolism induced by diverse types of cancer. UBR2 serves as the substrate recognition component of the N-end rule pathway [77]. Protein degradation via the N-end rule pathway accounts for a large portion of total protein ubiquitylation induced by such inflammatory conditions as sepsis, cancer, and diabetes [78,79,80]. UBR2 is the only known N-end rule pathway E3 that is upregulated by cachectic stimuli including tumor and pro-inflammatory cytokines (TNFα and IL-6), and is highly efficient for protein ubiquitylation via the N-end rule pathway [75], which makes it a potential candidate E3 that mediates myofibrillar protein loss through the UPP.

Utilizing mice with muscle-specific knockout of p38β MAPK, Liu et al. [81] demonstrated that p38β MAPK is critical to the coordinate activation of UPP and ALP leading to muscle wasting induced by LLC. Upon activation by p38β MAPK, C/EBPβ upregulates not only E3s atrogin1/MAFbx and UBR2, but also members of the Atg8 family, LC3b, and Gabarapl1. Surprisingly, p38β MAPK is also essential for the activation of Unc-51 Like Autophagy Activating Kinase 1 (ULK1) by phosphorylating its Ser-555 residue, which is required for the lipidation of the Atg8 family to form autophagosomes [82]. Contrary to the previous findings that adenosine monophosphate-activated protein kinase (AMPK) mediates the phosphorylation of ULK1 on Ser-555 that activates the kinase during nutrient deprivation [83,84], AMPK is not required for Ser-555 phosphorylation induced by LLC. These data uncover a central role of p38β MAPK in mediating cancer-induced muscle mass loss through orchestrating the activation of both the UPP and ALP (illustrated in Figure 1).

The p38β MAPK–C/EBPβ signaling is also important for the catabolic activity of some other catabolic factors found in the cancer milieu. Activation of ActRIIB by its agonists, activin A/B or myostatin, promotes muscle wasting [85]. Activins can be released by activated macrophages in response to TLR4 activation [86]. ActRIIB belongs to the TGFβ receptor superfamily and activates Smad2/3 signaling, which promotes muscle catabolism [87]. However, Smad3 knockout in mice does not spare the mice from muscle atrophy induced by the ActRIIB agonist myostatin [88], suggesting that an unidentified signaling mechanism is essential for ActRIIB-mediated muscle catabolism. Ding et al. [39] found that the catabolic effects of ActRIIB activation by activin A is abolished in mice with muscle-specific knockout of p38β MAPK. In addition to upregulating atrogin1/MAFbx and UBR2 through the p38β MAPK-C/EBPβ signaling pathway, activin A also activates autophagy in a p38β MAPK-dependent manner. On the other hand, activin A upregulates MuRF1 through a p38β MAPK-independent mechanism. Nevertheless, the increase in MuRF1 does not result in the loss of the myosin heavy chain in the absence of p38β MAPK, which supports the notion that the degradation of the myosin heavy chain in response to ActRIIB activation is mediated by an unidentified E3 that is upregulated by p38β MAPK. Conversely, ghrelin administration mitigated activation of p38 MAPK–C/EBPβ signaling and protected against muscle loss in LLC tumor-bearing mice [89].

Early evidence for a role of NF-κB in cachectic muscle wasting was shown by Li et al. in 1998 [41], where TNFα treatment of C2C12 myotubes rapidly activated NF-kB via stimulating the degradation of IκBα, resulting in increased protein ubiquitylation and myosin heavy chain loss. A subsequent study showed that TNFα upregulates the E2 ubiquitin-conjugating enzyme UbcH2 and increases protein ubiquitylation in C2C12 myotubes through the activation of NF-κB [42]. UbcH2 pairs with E3 ligase UBR2 to ubiquitylate specific substrates in chromatin-associated ubiquitylation [90], suggesting a coordination of NF-κB and p38β MAPK-mediated signaling in inflammation-associated skeletal muscle catabolism. In 2004, Cai et al. [91] demonstrated that NF-κB activation is sufficient to cause muscle mass loss, and is also required for LLC tumor-induced muscle wasting in mice through the upregulation of E3 ubiquitin ligase MuRF1. Corroborating data indicated that LPS upregulates MuRF1 in skeletal muscle in an NF-κB-dependent manner [92]. Another line of evidence, however, does not support the essential role of MuRF1 in cancer-induced muscle wasting. Upregulation of MuRF1 was not observed at various stages of cancer-induced muscle wasting in cell cultures [28], animal models [28,93], and cancer patients [26,94,95,96]. Given the high sensitivity of myosin to cancer-induced catabolism, it is conceivable that the loss of myosin could be mediated by a yet to be identified E3 in cancer cachexia. Furthermore, curcumin was shown to attenuate LPS-induced muscle catabolism while inhibiting the activation of p38 MAPK, but not NF-κB, suggesting p38 MAPK is a prominent mediator of muscle catabolism activated by TLR4 [97]. Thus, how NF-κB contributes to muscle catabolism in cancer cachexia remains to be defined.

In addition to mediating skeletal muscle catabolism, TLR4 signaling suppresses skeletal muscle regeneration by inhibiting myogenic differentiation through the activation of NF-κB [98]. Indeed, NF-κB mediates MyoD down-regulation induced by TNFα and interferon-γ, thereby contributes to skeletal muscle loss in cachexia by inhibiting muscle regeneration [99]. In addition, NF-κB inhibits skeletal myogenesis through upregulating YY1 [100] and Pax7 [101], leading to impaired transition from proliferation to terminal differentiation in myogenic progenitor cells.

As a transcription factor involved in numerous cellular processes, the signaling mechanisms that mediate C/EBPβ activation are highly complex [102,103]. In addition to its regulation by p38β MAPK through its phosphorylation on Thr-188 as discussed earlier, the most recent evidence [104] indicates that C/EBPβ activation in response to tumor requires a site-specific acetylation of its Lys-39 residue mediated by p300. Muscle specific knockout of p300 or pharmacological inhibition of p300 with C646 abrogates muscle wasting in LLC tumor-bearing mice. The novel data that recombinant Hsp70 and Hsp90 increase the activity of p300 suggest that p300 may act as a downstream effector of TLR4 in skeletal muscle. In addition, over-expression of a Lys-39 acetylation-defective C/EBPβ mutant protects against the LLC-induced muscle catabolism both in vitro and in vivo without affecting the Thr-188 phosphorylation status of C/EBPβ, indicating that p300-mediated acetylation of C/EBPβ plays a prominent role in the activation of this transcription factor in the cancer milieu.

## 5. Cachectic Cancers Induce Muscle Catabolism by Activating TLR4 through Releasing Hsp70 and Hsp90

To identify key cancer-released factors that induce muscle wasting, Zhang et al. [105] screened chromatographic fractions of LLC cell-conditioned medium and found that protein fractions with molecular weight of ~70 KD to ~90 KD accounted for the bulk of the catabolic activity in the medium. The active components in the fractions were identified as Hsp70 and Hsp90α/β. Structurally, both Hsp70 and Hsp90 have an N-terminal ATP-binding domain and a C-terminal substrate-binding domain with a linker in-between. The two prominent isoforms of Hsp90, Hsp90α and Hsp90β, are highly homologous with similar functions [106]. Under physiological conditions, Hsp70/90 and their cognates play essential roles in modulating protein–protein interactions, participating in the folding, assembly, and translocation of intracellular proteins [107,108]. When proteins are damaged beyond repair, Hsp70/90 target them for degradation by proteasomes [109,110]. Thus, Hsp70/90 are considered mainly intracellular molecular chaperones, which makes them cytoprotective against stressful stimuli. In skeletal muscle cells, overexpression of Hsp70 protects against muscle damage and age-related muscle dysfunction [111] as well as disuse or mechanical ventilation-induced muscle atrophy [112,113,114]. However, under pathological conditions, Hsp70 and Hsp90 can be released into extracellular space where they act as DAMPs [115]. Due to their unnatural growth properties, many tumor cells over-express Hsp70 and Hsp90 to maintain rapid metabolism. In addition, high levels of cell surface-bound Hsp70/90 are found in human malignant tumors including colorectal, lung, and pancreas carcinomas [116,117,118,119] and malignant melanoma/melanoma metastases [120]. These types of tumors are highly cachectic [121,122]. Such tumors release extracellular vesicles known as exosomes that display Hsp70 and Hsp90 on their surface [123,124].

Zhang and colleagues [105] found that Hsp70 and Hsp90 are constitutively released from all of the cachexia-inducing cancer cell lines examined at levels ~10-fold higher than non-tumorigenic cells and non-cachexia-inducing cancer cells. Importantly, in mice bearing LLC tumor or intestinal adenoma due to the Apc^min/+^ mutation, serum levels of Hsp70 and Hsp90 are elevated ~3-fold in sync with the development of cachexia. In contrast, serum levels of Hsp70 and Hsp90 in mice bearing EL4 tumor that do not develop cachexia remained unchanged. Furthermore, cancer cell-released Hsp70 and Hsp90 are associated with extracellular vesicles (EVs) that are characteristic of exosomes in terms of size, morphology, and protein markers. Tumor-bearing mice that received Hsp70/90-neutralizing antibodies or implanted with Hsp70/90-deficient tumor cells are resistant to the development of muscle wasting, despite the fact that the tumor growth is not affected. Blocking cancer release of EVs by silencing the expression of Rab27A/B also abrogates muscle wasting in LLC tumor-bearing mice. Conversely, administration of recombinant Hsp70 and Hsp90 recapitulates the catabolic effects of tumor in myotubes and tumor-free mice. They also demonstrated that isolated Hsp70/90-positive EVs induce myotube catabolism through the activation of TLR4 but not TLR2, supporting the notion that Hsp70/90 on the surface of EVs interact with and activate TLR4 on recipient cells specifically. Consequently, TLR4 knockout mice are spared from recombinant Hsp70 and Hsp90-induced muscle wasting. These discoveries depicted a new mechanism of cancer-induced muscle wasting as illustrated in Figure 2.

Moreover, this study showed that elevation of serum inflammatory cytokines (TNFα and IL-6) in tumor-bearing mice are in sync with and dependent on the elevation of serum Hsp70 and Hsp90. Thus, cancer cell-released Hsp70 and Hsp90 cause muscle wasting by activating TLR4 on muscle cells directly, and by activating TLR4 systemically to increase systemic inflammation indirectly. These findings identify cancer cell-released Hsp70 and Hsp90 as the primary drivers of muscle wasting in mouse models of cancer.

Although activation of TLR4 by LPS subsides quickly after continuous administration (tachyphylaxis), continuous administration of Hsp70/90 leads to muscle catabolism, simulating tumor-induced muscle without apparent tachyphylaxis [105]. The reason behind this difference is unknown. The possibility exists that this difference arises from the different manner through which Hsp70/90 interacts with TLR4 and co-receptor CD14. For example, LPS binding to LPS-binding protein (LBP) allows its transfer to CD14 through specific binding domains and stimulates CD14-dependent TLR4 internalization [125]. Due to sequence/structural differences from LPS, Hsp70/90 may not cause TLR4 internalization.

Patients with pancreatic cancer have the highest prevalence and severity of cachexia among all cancer patients [4]; however, the underlying mechanism is unknown. Yang and colleagues [126] found most recently that the high level expression of plasma membrane zinc transporter ZIP4 in human pancreatic cancer AsPC-1 and BxPC cell lines is responsible for their high cachexia-inducing capacity. ZIP4 promotes the release of EVs that carry Hsp70 and Hsp90 by pancreatic cancer cells through upregulating Rab27B, a small GTPase that controls EV release, resulting in a marked further elevation of circulating Hsp70 and Hsp90 that stimulate muscle catabolism.

Elevated serum Hsp70 and Hsp90 have been detected in patients with cachexia-prone cancers including lung [127,128,129,130,131,132], colorectal [133], and pancreatic cancer [134]. Importantly, serum Hsp70 and Hsp90 levels in cancer patients increase with the development of pathological grade and clinical stage [128,129,131], and the increase correlates with mortality in cancer patients [127,133]. Elevated EV levels have also been observed in the serum of lung, colorectal, and pancreatic cancer patients [135,136,137], which correlates with shorter survival [135]. These clinical data support a role of elevated circulating Hsp70 and Hsp90 in cancer cachexia. However, whether elevation in serum Hsp70/90 and EVs correlate to cancer cachexia and are causal to cancer cachexia in humans are yet to be established.

## 6. Conclusions

The above discussions depict a novel etiological paradigm of cancer-induced skeletal muscle wasting comprised of two parts. On the cancer part, the release of Hsp70/90-containing EVs extracts free amino acids from the body’s largest pool of proteins, skeletal muscle, perhaps to satisfy its need for rapid growth, by activating TLR4 systemically. On the muscle part, TLR4 stimulates muscle catabolism primarily by activating an intramuscular signaling cascade centered around p38β MAPK. In addition, through their own receptors, inflammatory cytokines and members of TGFβ superfamily also promote muscle catabolism by activating p38β MAPK-mediated signaling cascade. However, data supporting this paradigm are largely derived from cell culture and animal models of cancer cachexia. It is critical to verify the presence of this paradigm in cancer patients and determine whether targeting this paradigm effectively intervene in cancer cachexia. First and foremost, a causal relationship between the elevation of circulating Hsp70 and Hsp90 and cachexia in humans has not been established. If confirmed, etiology-based therapeutic strategies can be tested by targeting multiple signaling steps ranging from blocking the release of Hsp70 and Hsp90 by cancer cells, neutralizing the circulating Hsp70 and Hsp90, to inhibiting the activation of TLR4 or its downstream effectors p38β MAPK and p300 with pharmacological agents. In spite of the recent conceptual advances, there are many more mechanistic aspects of cancer cachexia that need to be addressed. For example, the significance of systemic activation of TLR4 by circulating Hsp70/90 in other tissues has not been defined. Although existing data may explain how muscle mass is lost, very little is known about the mechanism of fatigue associated with cachexia. Finally, how chemotherapeutic reagents promote cachexia is poorly understood. Continued efforts are required to solve these questions.

## Figures and Tables

**Figure 1 cancers-11-01272-f001:**
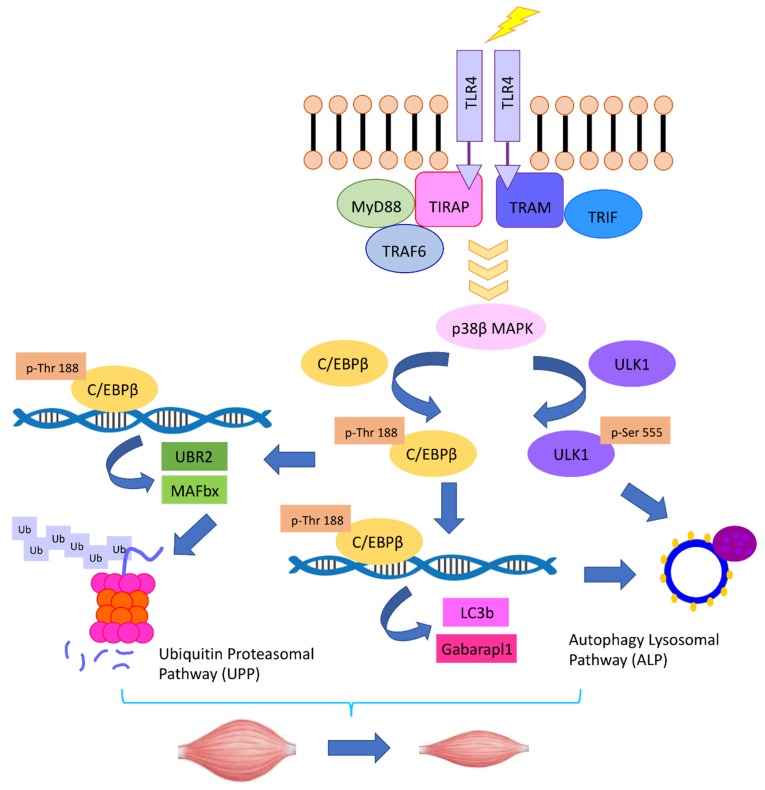
The TLR4–p38β MAPK–C/EBPβ signaling pathway mediates the coordinate activation of UPP and ALP in cancer-induced muscle catabolism.

**Figure 2 cancers-11-01272-f002:**
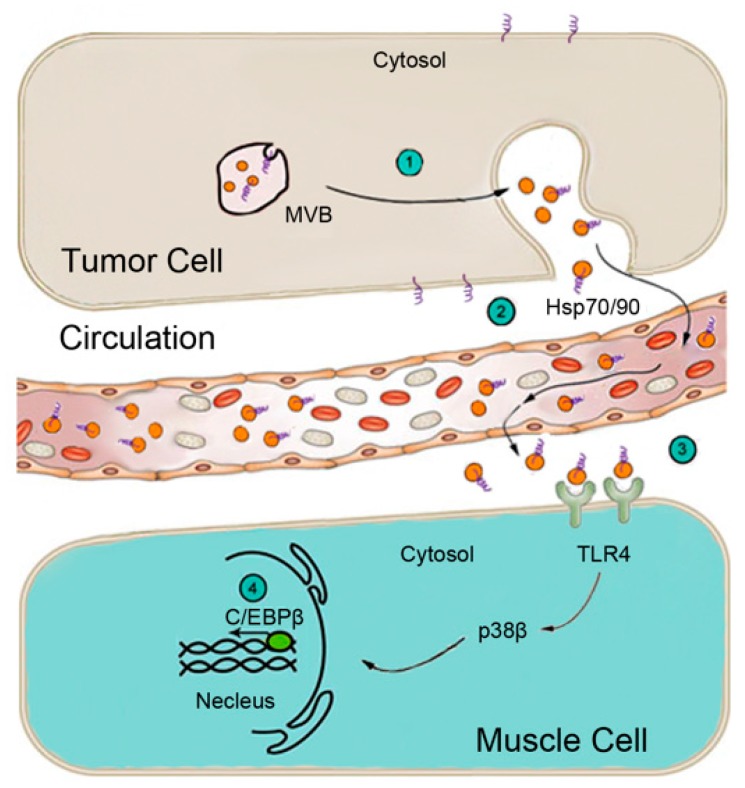
Schematic representation of the mechanism for cancer induction of muscle wasting through the release of EV-associated Hsp70 and Hsp90 that activate TLR4 on muscle cells. (1) Multivesicular bodies (MVBs) in tumor cells fuse with the plasma membrane releasing EVs (orange balls). (2) Circulating EVs carry surface Hsp70 and Hsp90 (purple loops). (3) Hsp70/90-carrying EVs activate TLR4 on muscle cells. (4) TLR4 activation of p38β MAPK upregulates rate-limiting genes in UPP and ALP via C/EBPβ.

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
