# Peer review of "Cancer Takes a Toll on Skeletal Muscle by Releasing Heat Shock Proteins—An Emerging Mechanism of Cancer-Induced Cachexia"

_cancers, 2019, doi:10.3390/cancers11091272_

Round 1

Reviewer 1 Report

Sin et al present a nice synthesis of what is known about TLR and related signaling in cancer cachexia and skeletal muscle more broadly. This work is well-referenced and makes a significant contribution to the cancer cachexia literature.  The illustrations are well done and significantly add to the readability of the manuscript. 
  The only addition I would suggest is a disclaimer than much of the cited mechanistic work, while robust, has been conducted in animals and that the field still has a long way to go to translating these mouse mechanisms to the clinic. This is very well done in lines 317 and 318 specifically about the HSP 70/90 work, but I feel it is at times unclear that the vast majority of the field’s mechanistic understanding comes from mouse work which may or may not hold up in cancer patients.   Minor issues of note: Line 114 appears to be missing a word after “vast”, most likely majority    Line 161 likely should be “On the other hand”, not “one”. There’s also an extra space on the line.

Author Response

We thank the reviewers for their very insightful comments.  Revisions have been made accordingly and highlighted in yellow.  Point to point responses are listed below.

Review 1

The only addition I would suggest is a disclaimer than much of the cited mechanistic work, while robust, has been conducted in animals and that the field still has a long way to go to translating these mouse mechanisms to the clinic. This is very well done in lines 317 and 318 specifically about the HSP 70/90 work, but I feel it is at times unclear that the vast majority of the field’s mechanistic understanding comes from mouse work which may or may not hold up in cancer patients.

R: This is a very important point.  We revised the abstract, the introduction and the conclusion to emphasize the need for verifications in humans.

Line 114 appears to be missing a word after “vast”, most likely majority 

R: Corrected.

Line 161 likely should be “On the other hand”, not “one”. There’s also an extra space on the line. 

R: Corrected.

Reviewer 2 Report

The review is very well-written and provides key information related to emerging mechanisms of cancer-induced cachexia. The authors performed an excellent synthesis of the literature, highlighting the main molecular mechanisms associated with this syndrome. The topic is both current and of interest to the readership of the journal. Although the manuscript is well written and flows seamlessly, there are few recommended corrections as outlined below.

Minor comments:

Line 35: Include the missing reference (~60% of all cancer patients); Lines 216-218: Revise the sentence for clarity; Line 243: Correct the typographic error: erliere > earlier.

Author Response

We thank the reviewers for their very insightful comments.  Revisions have been made accordingly and highlighted in yellow.  Point to point responses are listed below.

Review 2.

Line 35: Include the missing reference (~60% of all cancer patients); Lines 216-218: Revise the sentence for clarity; Line 243: Correct the typographic error: erliere > earlier.

R: All corrected.

Reviewer 3 Report

Overall a nice review, adding a new angle to the field that has not previously been covered. Well referenced throughout. I have some minor comments to recommend which include:

1) At time some evidence is cited without saying whether mouse/rat or human, or cell culture experiments were used to derive results. It would be beneficial if the authors could clarify this point throughout. 

2) L120 - Here there is a quick jump to TLR4. There is not mention of other members etc, so I think this needs addressing before moving on to discuss TLR4 specifically.

3) L162 - Is TLR4 expression also changed in other chronic disease such as heart failure and COPD? This would be very informative to add here. 

4) Fig.1 - Surely the TLR4 needs to be added in this figure, considering sall the discussion is about how this activates UPP and ALP.

5) I wonder what role disuse plays in activating TLR4? Is it likely cancer patients have less physical activity, so this may be one key factor that deserves some discussion.

6) The discussion of Hsp70/90 is insightful. What is interesting is that some studies have shown increase expression of Hsp72 prevents muscle (diaphragm) dysfunction (see Scott Powers group's work). This should be discussed to clarify why the Hsp here act in a different way to induce wasting. 

7) Are other muscle impacted by the Hsps? For example what happends to cardiac muscle - does this also waste?

8) It is vital a section addressing future directions is highlighted. Where are we now and where do we need to go next. This would be very useful for the reader to finish off a very high quality review.

Author Response

We thank the reviewers for their very insightful comments.  Revisions have been made accordingly and highlighted in yellow.  Point to point responses are listed below.

Review 3.

1) At time some evidence is cited without saying whether mouse/rat or human, or cell culture experiments were used to derive results. It would be beneficial if the authors could clarify this point throughout. 

R: We have added statements that the data covered by this review are mostly from lab models of cancer cachexia in Abstract and Introduction.  We also added clarifications in the text. 

2) L120 - Here there is a quick jump to TLR4. There is not mention of other members etc, so I think this needs addressing before moving on to discuss TLR4 specifically.

R: A new sentence has been added to bridge the transition to TLR4: “Indeed, emerging evidence has revealed the potential root cause of cancer-induced inflammation and cachexia”.  

3) L162 - Is TLR4 expression also changed in other chronic disease such as heart failure and COPD? This would be very informative to add here. 

R: We extended this paragraph by adding the implication of TLR4 in muscle wasting in cancer patients, heart failure and chronic kidney disease.  But we did not found information on TLR4 involvement in skeletal muscle of COPD.  Another paragraph has been added to cover TLR4 in cancer-induced loss of adipose tissue. 

4) Fig.1 - Surely the TLR4 needs to be added in this figure, considering sall the discussion is about how this activates UPP and ALP.

R: We have revised Figure 1. 

5) I wonder what role disuse plays in activating TLR4? Is it likely cancer patients have less physical activity, so this may be one key factor that deserves some discussion.

R: We have added a reference demonstrating that disuse-induced muscle atrophy is independent of TLR4 (Kawanishi et al., 2017), and disccussed its significance.

6) The discussion of Hsp70/90 is insightful. What is interesting is that some studies have shown increase expression of Hsp72 prevents muscle (diaphragm) dysfunction (see Scott Powers group's work). This should be discussed to clarify why the Hsp here act in a different way to induce wasting. 

R: Although intracellular HSPs are protective of muscle as molecular chaperons, extracellular HSPs are DAMPs causing inflammation as LPS. We have added some background info on this issue including Scott Powers group’s paper.

7) Are other muscle impacted by the Hsps? For example what happends to cardiac muscle - does this also waste?

R: We have not looked into that yet, although TLR4 is thought expressed by most muscles.  This is something needs be done in the future.

8) It is vital a section addressing future directions is highlighted. Where are we now and where do we need to go next. This would be very useful for the reader to finish off a very high quality review.

R: We have revised the Conclusion extensively to address future directions.